# Inhibiting NLRP3 Inflammasome Activation by CY-09 Helps to Restore Cerebral Glucose Metabolism in 3×Tg-AD Mice

**DOI:** 10.3390/antiox12030722

**Published:** 2023-03-15

**Authors:** Shuangxue Han, Zhijun He, Xia Hu, Xiaoqian Li, Kaixin Zheng, Yingying Huang, Peng Xiao, Qingguo Xie, Jiazuan Ni, Qiong Liu

**Affiliations:** 1Shenzhen Key Laboratory of Marine Biotechnology and Ecology, College of Life Sciences and Oceanography, Shenzhen University, Shenzhen 518055, China; 2College of Physics and Optoelectronic Engineering, Shenzhen University, Shenzhen 518060, China; 3College of Life Science and Technology, Huazhong University of Science and Technology, Wuhan 430074, China; 4Shenzhen Bay Laboratory, Shenzhen 518055, China; 5Shenzhen-Hong Kong Institute of Brain Science-Shenzhen Fundamental Research Institutions, Shenzhen 518055, China; 6National R&D Center for Se-Rich Agricultural Products Processing, Hubei Engineering Research Center for Deep Processing of Green Se-Rich Agricultural Products, School of Modern Industry for Selenium Science and Engineering, Wuhan Polytechnic University, Wuhan 430023, China; 7Department of Medical Physics and Biomedical Engineering, Istituto Neurologico Mediterraneo, Neuromed IRCCS, 86077 Pozzilli, Italy

**Keywords:** Alzheimer’s disease, glucose metabolism, NLRP3 inflammasome, insulin resistance, oxidative stress, CY-09

## Abstract

The reduction of the cerebral glucose metabolism is closely related to the activation of the NOD-like receptor protein 3 (NLRP3) inflammasome in Alzheimer’s disease (AD); however, its underlying mechanism remains unclear. In this paper, ^18^F-flurodeoxyglucose positron emission tomography was used to trace cerebral glucose metabolism in vivo, along with Western blotting and immunofluorescence assays to examine the expression and distribution of associated proteins. Glucose and insulin tolerance tests were carried out to detect insulin resistance, and the Morris water maze was used to test the spatial learning and memory ability of the mice. The results show increased NLRP3 inflammasome activation, elevated insulin resistance, and decreased glucose metabolism in 3×Tg-AD mice. Inhibiting NLRP3 inflammasome activation using CY-09, a specific inhibitor for NLRP3, may restore cerebral glucose metabolism by increasing the expression and distribution of glucose transporters and enzymes and attenuating insulin resistance in AD mice. Moreover, CY-09 helps to improve AD pathology and relieve cognitive impairment in these mice. Although CY-09 has no significant effect on ferroptosis, it can effectively reduce fatty acid synthesis and lipid peroxidation. These findings provide new evidence for NLRP3 inflammasome as a therapeutic target for AD, suggesting that CY-09 may be a potential drug for the treatment of this disease.

## 1. Introduction

Alzheimer’s disease (AD) is a common neurodegenerative disease in elder people. Deposits of amyloid-β (Aβ) and hyperphosphorylated tau protein are its main characteristics [1,2]. Due to the failure of anti-amyloid and anti-tau aggregation drugs, neuroinflammation has been considered as a new therapeutic target for AD treatment.

NLRP3 inflammasome, composed of nucleotide-binding oligomerization domain (NOD)-like receptor protein 3 (NLRP3) and apoptosis-associated speck-like proteins of CARD (caspase recruitment domain) (ASC) and pro-caspase-1, plays an important role in neuroinflammation. Activation of NLRP3 inflammasome causes the increase in caspase-1 and the release of interleukin-1β (IL-1β) [3,4]. Increased activation of NLRP3 inflammasome is closely related with the reduction of cerebral glucose metabolism. Studies show that the translocation of hexokinase (HK), a key enzyme in glucose metabolism, can activate NLRP3 inflammasome, and inhibition of NLRP3 inflammasome can restore the expression and distribution of HK in AD model cells [5,6]. In AD mice, NLRP3 binds to the mitochondria and is then activated by mitochondrial reactive oxygen species (mtROS) [7,8]. HK binding to mitochondria reduces mtROS transport into the cytoplasm and further reduces the activation of NLRP3 inflammasome [9,10]. A positive correlation occurs between glucose metabolism and neuroinflammation in early AD, but disappears as the pathological course progresses [11]. In addition, increased neuroinflammation leads to a shift in energy metabolism from oxidative phosphorylation (OxPhos) to aerobic glycolysis in microglia. Activated microglia compete with neurons for glucose, which limits the energy availability of neurons [12]. Our previous studies have shown that inhibiting the activation of NLRP3 inflammasome can increase the expression and distribution of HK in vitro [5], but determining whether it could restore cerebral glucose metabolism in this process requires further exploration in vivo.

Glucose metabolism provides over 70% of energy for the brain. Glucose is transported into cells via glucose transporters (GLUTs), is phosphorylated by HK, and then catalyzed by a series of enzymes to produce energy. GLUT1 and GLUT3 are expressed in the brain and are responsible for the entry of glucose from the blood to the intracellular environment [13,14,15]. In AD, decreased expression levels of GLUT1 and GLUT3 lead to reduced glucose transport. GLUT4 is also expressed in the brain. Unlike GLUT1 and GLUT3, it is sensitive to insulin, and when insulin levels rise, it is transferred to cell membranes for the transport of glucose, [16,17]. The transfer of GLUT4 is regulated by the insulin-PI3K-AKT pathway [18]. Several studies have shown that insulin resistance is a key event leading to AD pathology [17,19,20]. Insulin receptors (IR) recognize insulin and then self-phosphorylate to recruit insulin receptor substrate (IRS) and activate the IRS-AKT-AS160 pathway [21,22]. AS160 is a guanosine triphosphate enzyme activating protein, the phosphorylation of which promotes translocation of GLUT4 [18,23]. In AD, lower insulin levels in cerebrospinal fluid result in decreased expression of p-IR, p-AKT, and p-AS160, a s well as increased IR [24]. A positron emission tomography (PET) study confirmed that the expression of IR was increased in the lower glucose metabolism region.

Oxidative stress also contributes to the pathology of AD. The generation of an excessive amount of ROS results in oxidative stress and the activation of NLRP3 inflammasome. Moreover, oxidative stress causes iron metabolism disorders and leads to ferroptosis [25,26]. Ferroptosis manifests as increased cellular Fe^2+^, decreased glutathione peroxidase 4 (GPX4), and increased lipid peroxidation [27]. Elevated Fe^2+^ results from increased transferrin transport and ferritinophagy [28]. Transferrin (TF), transferrin receptor (TFR), and ferroportin (FPN) are responsible for the transport of Fe, and nuclear receptor coactivator 4 (NCOA4) participates in ferritinophagy. Long-chain acyl-CoA synthetase 4 (ACSL4) is a key protein linking ferroptosis and lipid peroxidation. Increased ACSL4 is consistent with increased ferroptosis and lipid peroxidation in AD [29,30,31]. Decreased GPX4 and Solute Carrier Family 7, Member 11 (SLC7A11) are also related to lipid peroxidation.

In the study, we injected CY-09, a specific inhibitor of NLRP3, into non-transgenic (NTg) and triple transgenic AD (3×Tg-AD) mice daily, with a dose of 2.5 mg/kg for six weeks, according to the reference [32]. Then, we explored the effect of NLRP3 inflammasome inactivation on glucose transport, insulin resistance, and glucose metabolic enzymes in vivo. Simultaneously, we investigated whether NLRP3 inflammasome inactivation by CY-09 could reduce AD classical pathology and oxidative stress and improve cognitive deficits. Overall, this study aimed to determine the effect of NLRP3 inflammasome activation on glucose metabolism and to investigate the potentiality of CY-09 as a therapeutic drug for AD treatment.

## 2. Materials and Methods

### 2.1. Reagents and Antibodies

CY-09 (Cat#S5774) was purchased from Selleck Chemicals LLC, Houston, TX, USA. Polyoxyl 15 hydroxystearate (Cat#HY-136349) and DMSO (Cat#HY-Y0320) were purchased from MedChemExpress LLC, Deer Park Dr, Suite Q, Monmouth Junction, NJ, USA. Saline (Cat#R22172) was purchased from Shanghai yuanye Bio-Technology Co., Ltd., Shanghai, China. ^18^F-FDG was provided by Union Hospital, Tongji Medical College, Huazhong University of Science and Technology, Wuhan, China. Antibodies against NLRP3 (Cat#AG-20B-0014-C100) and caspase-1 (P20) (Cat#AG-20B-0042-C100) were purchased from Adipogen Corporation, San Diego, CA, USA. Antibodies against IL-1β (Cat#16806-1-AP), GLUT1 (Cat#21829-1-AP), GLUT4 (Cat#66846-1-Ig), GLUT3 (Cat#20403-1-AP), HK2 (Cat#22029-1-AP), voltage-dependent anion-selective channel protein 1 (VDAC1) (Cat#10866-1-AP), and SLC7A11 (Cat#26864-1-AP) were purchased from Proteintech Group, Inc., Chicago, IL, USA. Antibodies against IRS (Cat#3407), p-IRS-Ser1101 (Cat#2385), AKT (Cat#9272), p-AKT-Ser473 (Cat#4060), GSK3β (Cat#12456), p-GSK3β-Ser9 (Cat#5558), FAS (Cat#3180), ACC (Cat#3676), p-ACC-Ser79 (Cat#11818), and LRP1 (Cat#64099) were purchased from Cell Signaling Technology, Inc., Boston, MA, USA. Antibodies against IR (Cat#sc-57342), p-IR-Tyr1150 (Cat#sc-81500), and ACSL4 (Cat#sc-271800) were purchased from Santa Cruz Biotechnology, Inc., Dallas, TX, USA. The antibodies against AS160 (Cat#ab189890), p-AS160-T642 (Cat#ab131214), HK1 (Cat#ab150423), PDHE1α (Cat#ab168379), COX IV (Cat#ab16056), APP (Cat#ab32136), BACE1 (Cat#ab108394), PSD95 (Cat#ab18258), synaptophysin (Cat#ab32127), tau5 (Cat#ab80579), p-tau-Ser404 (Cat#ab92676), MDA (Cat#ab27642), GPX4 (Cat#ab125066), and HMGCS1 (Cat#ab155787) were purchased from Abcam plc., Cambridge, UK. Antibodies against 6E10 (Cat#803002), sAPPα (Cat#813501), and sAPPβ (Cat#813401) were purchased from BioLegend, Inc., San Diego, CA, USA. The antibody against Aβ1-42 (Cat#AB5078P) was purchased from Millipore Corporation, Boston, MA, USA. Antibodies against HT7 (Cat#MN1000), TF (Cat#PA5-27306), TFR (Cat#13-6800), FPN (Cat#PA5-22993), NOCA4 (Cat#PA5-96398), and SREBP2 (Cat#PA1-338) were purchased from Thermo Fisher Scientific, Waltham, MA, USA. The antibody against β-actin (Cat#AB0033) was purchased from Abways Technology, Inc., Shanghai, China. The antibody against HMGCR (Cat#T56640S) was purchased from Abmart Shanghai Co., Ltd., Shanghai, China. Goat Anti-Rabbit IgG H&L (HRP) secondary antibody (Cat#ab6721) and Goat Anti-Mouse IgG H&L (HRP) secondary antibody (Cat#ab6789) were purchased from Abcam plc., Cambridge, UK. Alexa Fluor^®^ 488 AffiniPure Goat Anti-Rabbit IgG (H+L) secondary antibody (Cat#111-545-003) and Alexa Fluor^®^ 594 AffiniPure Goat Anti-Mouse IgG (H+L) secondary antibody (Cat#111-585-003) were purchased from Jackson ImmunoResearch Inc., West Grove, PA, USA. DAPI Staining Solution (Cat#C1002) was purchased from Beyotime Biotech Inc., Shanghai, China.

### 2.2. Animals and Treatment

The impact of sex on AD pathology has been reported in many references. Senior females are more likely to develop AD due to their lower estrogen levels. Ovarian hormone loss causes a bioenergetic deficit and a shift in metabolic fuel availability in AD model mice [33,34]. Thus, we select 9-month-old female C57BL/6J mice (NTg mice) and 3×Tg-AD mice for this study. NTg mice were purchased from Guangdong Medical Laboratory Animal Center and 3×Tg-AD mice which harbor the mutated human genes amyloid precursor protein (APP) (SWE), PS1 (M146V), and Tau (P301L) were purchased from the Jackson Laboratory. Mice were housed in conditions under 22 °C in a 12:12 h light/dark cycle, with food and water ad libitum. All animal experimental protocols were reviewed and approved by the Institutional Animal Care and Use Committee of Tongji Medical College of Huazhong University of Science and Technology (IACUC Number: 2390; Approved Date: 27 February 2018).

To observe the impact of NLRP3 inflammasome activation on cerebral glucose metabolism in AD, we used CY-09 to inhibit NLRP3 inflammasome activation. Here, 24 female mice were randomly divided into four groups: NTg mice, CY-09-treated NTg mice (NTg + CY-09 mice), 3×Tg-AD mice, and CY-09-treated 3×Tg-AD mice (3×Tg-AD + CY-09 mice). CY-09 was dissolved in a vehicle containing 10% DMSO, 10% Polyoxyl 15 hydroxystearate, and 80% saline. It was then injected into the mice daily, with a dose of 2.5 mg/kg for six weeks, according to the reference [32].

### 2.3. ^18^F-FDG PET

^18^F-FDG PET experiments were applied to examine the glucose metabolism in the mouse brain [35]. All the mice were fasted for 12–16 h and then injected by vein with 7.4 MBq ^18^F-FDG before the PET scan. For the static PET scan, the mice were scanned for 10 min after 60 min of free metabolism. However, in the dynamic scan, the mice were scanned for 60 min immediately after injection. PET data were acquired with Trans-PET Discoverist 180 (Raycan Technology Co., Ltd., Suzhou, China) and reconstructed with a 3D OSEM algorithm. The time segmentary schemes for dynamic data reconstruction were as follows: 5 s × 6, 10 s × 3, 30 s × 4, 60 s × 2, 120 s × 5, 300 s × 3, 600 s × 1, 300 s × 1, 900 s × 1. To calculate the cerebral metabolic rate of glucose (CMRglu), blood glucose was measured from the second drop of tail blood using a Roche blood glucose meter (Roche Pharma (Schweiz) Ltd., Basel, Switzerland). Standard uptake values (SUVs) of the whole brain, cortex, and hippocampus, as well as the CMRglu of the whole brain, were quantified by Amide 1.0.4 software (Crump Institute for Molecular Imaging, UCLA School of Medicine, CA, USA), Crimas 2.9 (Turku PET center, Turku, Finland), MATLAB 2019b (MathWorks Inc., Natick, MA, USA), and SPM12 (The Wellcome Centre for Human Neuroimaging, UCL Queen Square Institute of Neurology, London, UK).

### 2.4. Morris Water Maze Test

The Morris water maze test is one of the behavioral experiments used to detect the spatial learning and memory ability of mice. In the study, we mainly refer to the previous protocol, with slight modifications [36,37]. None of the mice were trained on the test before experimentation. However, an additional process was carried out to familiarize the mice with the environment before the formal experiment started. The Morris water maze test includes place navigation tests and spatial probe tests. In the place navigation test, the mice were first placed on the platform for 1 min to familiarize themselves with the environment. Subsequently, the mice were placed into the water from the four quadrants. The time (escape latency) from entering the water to finding the platform was recorded within 60 s. The place navigation test lasted for five days. In the spatial probe test, the platform was removed and the mice were placed into the diagonal quadrant. Then, the time spent by the mice in the target quadrant in 24 h and 72 h was recorded. The Morris water maze WMT-200A (Chengdu Techman Software Co., Ltd., Chengdu, China) and the Animal Behavior Analysis System BAS-100 (Chengdu Techman Software Co., Ltd., Chengdu, China) were used for the recording of data.

### 2.5. Glucose Tolerance Test (GTT) and Insulin Tolerance Test (ITT)

GTT and ITT were performed to respectively detect the ability to regulate blood glucose and insulin sensitivity [32,38]. In GTT, the mice were fasted for 12 h and then injected with glucose in the dose of 2 g glucose per Kg mice (glucose was purchased from Beyotime Biotech Inc., Shanghai, China, Cat#ST1228). In ITT, mice were fasted for 4 h and then injected with insulin in the dose of 0.75 U insulin per Kg mice (insulin was purchased from Beyotime Biotech Inc., Shanghai, China, Cat#P3376). All experiments were required to detect and record the blood glucose in time points of 0, 30, 60, 90, 120, and 150 min after the injection. Blood glucose was measured from the second drop of tail blood using a Roche blood glucose meter (Roche Pharma (Schweiz) Ltd., Basel, Switzerland).

### 2.6. Brain Tissues Extraction and Preservation

After a week of recovery, the mice were euthanized after isoflurane anesthesia to extract the brain and blood. The hemibrain of each mouse was used for immunofluorescence experiments and stored with 4% paraformaldehyde. The total proteins of the rest of the hippocampus were extracted using the Phosphorylated Protein Extraction Kit (Cat#KGP950, Jiangsu Keygen Biotech Corp., Ltd., Nanjing, China) for Western blot experiments. The of the rest cortex was used for LC-MS/MS analysis, ROS measurement, and mitochondrion isolation. All tissues were stored at −80 °C before the experiments. Blood was tested for insulin directly after the extraction.

### 2.7. ELISA Assay and Reactive Oxygen Species Measurement

All experiments were conducted following the instructions of each kit. Fasting blood insulin was detected using a Highly Sensitive Mouse Insulin Immunoassay Kit (Cat#HMS200, EZassay Ltd., Shenzhen, China). The secretion of IL-1β was detected using a Mouse IL-1β ELISA Kit (Cat#EMC001b, Neobioscience Technology Company, Shenzhen, China). ROS levels were measured using the Reactive Oxygen Species Assay Kit (Cat#E004-1-1, Nanjing Jiancheng Bioengineering Institute, Nanjing, China). These data were detected by a SpectraMax^®^ L Microplate Reader (Molecular Devices, LLC, San Jose, CA, USA).

### 2.8. LC-MS/MS

LC-MS/MS was performed by Triple TOF 6500 (AB Sciex LLC, Framingham, MA, USA) to determine whether CY-09 crosses the BBB and enters the brains of the mice. In this experiment, metabolites were extracted from brain tissues using a 300 μL methanol acetonitrile mixture (methanol: acetonitrile = 2:1), as previously described [39]. After vortexing for 1 min, sonicating for 10 min at 0 °C, and centrifuging for 15 min at 13,000× *g*, the samples were held at −20 °C until further detection. The conditions for the chromatographic separation of metabolites were as follows—flow rate: 0.3 mL/min; injection volume: 6 μL; mobile phase: phase A, water (containing 0.1% formic acid); phase B, acetonitrile (containing 0.1% formic acid). The ion source parameters of mass spectrometry were as follows—curtain gas: 35 psi; ion spray voltage: −4500 V; source temperature: 550 °C; ion source gas1: 55psi; ion source gas2: 55psi. MultiQuant 3.02 (AB Sciex LLC, Framingham, MA, USA) and ProteoWizard 1.3.5.0 (ProteoWizard, Palo Alto, CA, USA) were used to process the data.

### 2.9. Mitochondrion Isolation and Hexokinase Activity

The brain mitochondria were isolated following the product instructions for the mitochondrial isolation kit (Cat#C3606, Beyotime Biotech Inc, Shanghai, China). Isolated mitochondria and cytoplasm were collected and stored at −80 °C. A NanoDrop 2000c (Thermo Fisher Scientific, Waltham, MA, USA) was used to detect the mitochondria concentration. The hexokinase activity was measured by Micro Hexokinase Assay Kit (Cat#BC0745, Beijing Solarbio Science & Technology Co., Ltd., Beijing, China). All experiments were carried out according to the product instructions.

### 2.10. Western Blot Analysis

The total proteins of the brain tissues were extracted using the Phosphorylated Protein Extraction Kit (Cat#KGP950, Jiangsu Keygen Biotech Corp., Ltd., Nanjing, China). The PierceTM BCA Protein Assay Kit (Cat#23225, Thermo Fisher Scientific, Waltham, MA, USA) was used for the detection of protein concentrations. Experimental protocols were the same as previously described [5]. In short, 10% and 12% sodium dodecyl sulfate polyacrylamide gel electrophoresis (SDS-PAGE) gels were used in the experiments. Proteins larger than 100 kD were separated using a 10% SDS-PAGE gel, while proteins smaller than 100 kD were separated using a 12% SDS-PAGE gel. Equal amounts of total protein lysates (20 μg per well) of each sample were loaded for the electrophoresis. Then, the proteins were transferred to a 0.45 μm polyvinylidene difluoride (PVDF) membrane (Cat#ISEQ00010, Millipore Corporation, Boston, MA, USA). After the electrotransfer, the membranes were blocked in 5% non-fat milk for 2 h and washed four times (10 min each time) with TBST buffer (150 mM NaCl, 10 mM Tris, 0.1% Tween-20, pH 7.4). Afterwards, the membranes were incubated with primary antibodies overnight at 4 °C. The next day, the membranes were incubated with secondary antibodies at room temperature for 2 h after the washing of the primary antibody with TBST buffer. After the washing of the secondary antibodies, immunoreactive bands were visualized by the Tanon 5200 Series Image Analysis System (Tanon Science & Technology Co., Ltd., Shanghai, China). Grayscale analysis was used to quantify the protein expression. Grayscale values of the target protein were normalized to the grayscale values of β-actin or VDAC1. ImageJ 1.53C (NIH, Bethesda, MD, USA) was used to quantify the grayscale values of each immunoreactive band.

### 2.11. Immunofluorescence Assay

An immunofluorescence assay was performed for the brain slices following the previously established protocol [40]. Briefly, the brain slices were dewaxed and the xylene removed with dimethylbenzene and different concentrations of alcohol (100%, 95%, 85%, 70%, 50%, 30%, and 0%), respectively. Next, the brain slices were incubated with primary antibodies overnight after antigen retrieval, membrane rupture, and blocking. The next day, brain slices were washed using PBST for 30 min and then incubated with a second antibody for 2 h in the dark. After the washing of the second antibody, the brain slices were placed on slides and observed using an Olympus BX53 microscope (Olympus Corporation, Tokyo, Japan).

### 2.12. Statistical Analysis

In this study, semi-quantitative analysis and absolute quantification analysis were used to quantify the SUVs and the CMR_glu_, respectively [41]. The equation for *SUV* is as follows:(1)$$SUV=\frac{C_{T}}{D_{Inj}}\cdot W_{s}$$

*C_T_*, *D_Inj_*, and *W_S_* represent the radioactivity in the region of interest, the injected dose of radioactivity, and the weight of the mouse, respectively.

The simplified equation for *CMR_glu_* is as follows:(2)$${CMR}_{glu}=\frac{C_{g}}{LC}\cdot K$$

*Cg* is the concentration of blood glucose. *LC* is the lumped constant, and it reflects the difference between the metabolism of ^18^F-FDG and glucose. *K* is the uptake rate of ^18^F-FDG.

All data were presented as mean ± SD and analyzed with GraphPad Prism 9 (GraphPad Software Inc., La Jolla, CA, USA). The normality of distribution of the results was checked by Shapiro–Wilk test. A normal distribution of the data was indicated by the test; thus, a one-way ANOVA, followed by Bonferroni’s multiple comparisons test, were performed in the study. Every possible comparison was explored, and significant differences (*p* < 0.05) between each group were shown in the figures.

## 3. Results

### 3.1. CY-09 Could Cross the Blood-Brain Barrier In Vivo

CY-09 inhibits the assembly and activation of the NLRP3 inflammasome by its combination with the ATP-binding motif of the NLRP3 NACHT domain to inhibit the ATPase activity. To investigate whether it is possible for CY-09 to cross the blood–brain barrier (BBB) in mice, initially, we adopted LC-MS/MS to detect the content of CY-09 in the brain of each group, i.e., the NTg, NTg + CY-09, 3×Tg-AD, and 3×Tg-AD + CY-09 mice. The results were shown in Figure 1a and Appendix A; CY-09 was found in the brain tissues of NTg + CY-09 and 3×Tg-AD + CY-09 mice, which indicated that CY-09 could cross the BBB in vivo.

### 3.2. CY-09 Inhibited NLRP3 Inflammasome Activation in 3×Tg-AD Mice

Then, Western blot and ELISA were then used to detect the expressions of NLRP3 inflammasome-related proteins to confirm the effect of CY-09 on NLRP3 inflammasome activation. As demonstrated in Figure 1b–h, compared with NTg mice, the expressions of NLRP3, pro-caspase-1, caspase-1(P20), and IL-1β were significantly increased in 3×Tg-AD mice, with the *p*-values lower than 0.05 and 0.01 respectively. Simultaneously, a notable reduction in the expressions of these proteins was found in CY-09 treated 3×Tg-AD mice, with the *p*-values lower than 0.05 and 0.01, respectively. Moreover, the expressions of pro-caspase-1 and caspase-1 were remarkably decreased in NTg + CY-09 mice, with the *p*-values all lower than 0.01. Except for a significant difference between NTg mice, NTg + CY-09 mice and 3×Tg-AD + CY-09 mice, the results of IL-1β secretion measured by ELISA were consistent with those of Western blotting. No differences were found in the expression of pro-IL-1β among the four groups of mice. Together, these data suggested that CY-09 has an inhibiting effect on NLRP3 inflammasome activation in the brain of triple transgenic AD mice.

### 3.3. CY-09 Increased Cerebral Glucose Metabolism in 3×Tg-AD Mice

Next, we evaluated the effect of NLRP3 inflammasome activation on cerebral glucose metabolism using static and dynamic PET. ^18^F-FDG is the most commonly used PET tracer for glucose metabolism. As an analog of glucose, ^18^F-FDG is transported into cells by GLUTs from blood after the i.v. injection and is then phosphorylated by HK. Due to the differences in structure, 6-P-^18^F-FDG cannot be catalyzed by glucose-6-phosphate isomerase and must remain in the cytoplasm. The amount and distribution of 6-P-^18^F-FDG represent the glucose metabolism levels in different brain regions.

Static PET results are shown in Figure 2a–e; compared with NTg mice, the standard uptake values (SUVs) of the whole brain, the cortex, and the hippocampus were greatly decreased in 3×Tg-AD mice, with the *p*-value lower than 0.05. After CY-09 treatment, the SUVs were significantly higher in 3×Tg-AD mice than those in non-treated AD mice, with a *p*-value lower than 0.05. There were no differences in the weight of the mice in the four groups. Consistent with the static PET results, the cerebral SUVs of NTg and CY-09 treated 3×Tg-AD mice were also higher than those of the 3×Tg-AD mice, even though the SUVs of the four groups of mice increased in dynamic PET over time (Figure 2f–h). Moreover, the results of the cerebral metabolic rate of glucose (CMRglu) showed a notable reduction in the 3×Tg-AD mice and a remarkable increase in the CY-09 treated 3×Tg-AD mice, with the *p*-values all lower than 0.05. Overall, the PET data demonstrated that inhibiting NLRP3 inflammasome activation helps to restore cerebral glucose metabolism in the 3×Tg-AD mice.

### 3.4. CY-09 Increased Glucose Transport in 3×Tg-AD Mice

GLUTs are responsible for glucose transport, which is the basis of glucose metabolism. GLUT1, GLUT3, and GLUT4 can be expressed in the brain. Here, we used Western blotting to analyze the expression of GLUT1, GLUT3, and GLUT4. As shown in Figure 3a–d, the expression of GLUT1, GLUT3, and GLUT4 were lower in 3×Tg-AD mice than in NTg mice, with the *p*-value lower than 0.01, 0.05, and 0.01 respectively. However, the expressions of GLUT1 and GLUT4 were significantly increased in CY-09 treated 3×Tg-AD mice than in non-treated AD mice, with the *p*-value lower than 0.05. The expression of GLUT3 was increased with a *p*-value of 0.077. Further, immunostaining of GLUT4 demonstrated the decreased distribution in 3×Tg-AD mice and the increased distribution in CY-09 treated 3×Tg-AD mice (Figure 3e and Appendix A). Hence, these data exhibited that inhibiting NLRP3 inflammasome activation can increase the expression and distribution of GLUTs in the 3×Tg-AD mice.

### 3.5. CY-09 Attenuated Insulin Resistance in 3×Tg-AD Mice

GLUT4 is regulated by the insulin signaling pathway to participate in glucose metabolism. Insulin resistance manifests itself as insensitivity to insulin and a higher level of insulin and glucose in the blood, eventually leading to impaired insulin signaling pathways and glucose metabolism. It had been confirmed to exist in AD.

To evaluate the effect of NLRP3 inflammasome activation on insulin resistance in AD, we first detected blood glucose and blood insulin levels. As shown in Figure 4, higher fasting and basal blood glucose and fasting insulin level were found in 3×Tg-AD mice compared with NTg mice, with the *p*-values all lower than 0.05 and 0.01, respectively. Results of the GTT and ITT showed an increased glucose tolerance and decreased insulin tolerance in 3×Tg-AD mice. However, after CY-09 treatment, blood glucose and insulin levels were decreased and better insulin sensitivity was found in CY-09-treated 3×Tg-AD mice. Thus, the results indicated that inhibiting NLRP3 inflammasome activation can attenuate insulin resistance in the 3×Tg-AD mice.

Then, to further explore the underlying mechanism by which NLRP3 inflammasome activation affects insulin resistance, we detected the expression and distribution of the IR-AKT-AS160 insulin signaling pathway-related proteins in NTg, NTg + CY-09, 3×Tg-AD, and 3×Tg-AD + CY-09 mice. As shown in Figure 5a,b, compared with NTg mice, expression of p-IR-Tyr1150 (phosphorylation protein of IR at Tyr1150) relative to IR was significantly decreased in 3×Tg-AD mice, with a *p*-value lower than 0.01, but the expression was increased after the CY-09 treatment, with a *p*-value lower than 0.05. The distribution of p-IR-Tyr1150 was found to be lower in the 3×Tg-AD mice and higher in the 3×Tg-AD + CY-09 mice. By contrast, the expression and distribution of IR were not different among the four groups of mice (Figure 5i and Appendix A). These findings suggested that inhibiting NLRP3 inflammasome activation can enhance the self-phosphorylation of IR in the 3×Tg-AD mice.

Self-phosphorylation of IR recruits IRS and starts the IRS-AKT-AS160 insulin signaling pathway. In parallel, we detected the expression of IRS, AKT, and AS160 and their phosphorylated levels in the four groups of mice. The results reported in Figure 5c–h show a notable reduction of p-AKT-Ser473 and p-AS160-T642, while significantly increased p-IRS-Ser1101 was found in the 3×Tg-AD mice, with the *p*-value lower than 0.01, 0.01, and 0.05, respectively. Treatment with CY-09 helped to reverse the expression of these proteins in 3×Tg-AD + CY-09 mice. There was also a significant reduction in IRS between the NTg + CY-09 mice and 3×Tg-AD + CY-09 mice, with a *p*-value less than 0.01. No differences were found in the expressions of AKT and AS160 among the four groups of mice. All the results showed that inhibiting NLRP3 inflammasome activation can restore the IR-IRS-AKT-AS160 insulin signaling pathway to alleviate insulin resistance in the 3×Tg-AD mice.

### 3.6. CY-09 Increased the Expression and Distribution of Metabolic Enzymes in 3×Tg-AD Mice

Increased glucose transport and improved insulin resistance were found in CY-09-treated 3×Tg-AD mice; we also detected the expressions and distribution of HK, which is the first enzyme that phosphorylates glucose when associated with VDAC1 in the mitochondria. We previously reported that the expression of cHK1 (HK1 in the cytoplasm) was significantly increased in 3×Tg-AD mice, while the expression of mHK1 (HK1 in the mitochondria) was remarkably decreased. However, increased HK1 expression and HK activity were found in CY-09-treated N2a-sw cells (a model cell of AD). These results prompted us to examine the expression and activity of HK in CY-09-treated 3×Tg-AD mice. Here, we isolated the mitochondria and cytoplasm from NTg, NTg + CY-09, 3×Tg-AD, and 3×Tg-AD + CY-09 mice to detect the expression and distribution of HK1.

As shown in Figure 6, consistent with our previous results, increased cHK1 and significantly decreased mHK1 and mHK2 were found in the 3×Tg-AD mice, with the *p*-value lower than 0.01 and 0.05. However, the expression of mHK1 was increased, while cHK1 was decreased in the CY-09-treated 3×Tg-AD mice, and the differences were significant compared to the untreated 3×Tg-AD mice, with the *p*-values lower than 0.01 and 0.05. cHK2 was also found to be notably decreased in the CY-09-treated 3×Tg-AD mice, with a *p*-value lower than 0.05. Although HK activity was remarkably decreased in 3×Tg-AD + CY-09 mice than in NTg + CY-09 mice, it was notably increased in the CY-09-treated 3×Tg-AD mice compared to the non-treated 3×Tg-AD mice, with a *p*-value lower than 0.05 (Figure 6b). Besides, we also detected the expressions of pyruvate dehydrogenase α 1 (PDHE1α) and cytochrome c oxidase subunit IV (COX4). They were all significantly reduced in 3×Tg-AD mice, with *p*-values lower than 0.05. Meanwhile, the expression of PDHE1α was remarkably increased in CY-09-treated 3×Tg-AD mice. In short, the data demonstrated an increase in the expression and distribution of HK by inhibiting NLRP3 inflammasome activation in the 3×Tg-AD mice.

### 3.7. CY-09 Relieved Cognitive Impairment and Pathological Injury in 3×Tg-AD Mice

Previous studies showed that NLRP3 inflammasome activation contributes to the pathology of AD. The results of this study demonstrated that inhibition of the NLRP3 inflammasome by CY-09 significantly recovered glucose metabolism. Next, we focused on the effects of CY-09 on cognitive impairment and classically pathological biomarkers in AD and confirmed whether CY-09 is a potential therapeutic drug for AD.

The Morris water maze was used to evaluate the learning and memory abilities of the four groups of mice. As shown in Figure 7, on the fifth day, the escape latency of 3×Tg-AD mice was remarkably longer than that of the NTg mice, but notably reduced after CY-09 treatment, with the *p*-values all lower than 0.05. Moreover, the time in the target quadrants of the 3×Tg-AD mice was less than that of the NTg mice, while increasing after CY-09 treatment in 24 h and 72 h space exploration experiments, with a *p*-value lower than 0.05 and a *p*-value of 0.1282, respectively. No differences were found in swimming speed between the four groups of mice. The results exhibited that inhibition of the NLRP3 inflammasome by CY-09 helped to relieve the cognitive impairment of the 3×Tg-AD mice.

Afterward, we detected the expression and distribution of pathological proteins in AD model mice. As shown in Figure 8, compared with the NTg mice, expressions of APP, beta-site app cleaving enzyme 1 (BACE1), sAPPβ, and Aβ1-42 were significantly increased in 3×Tg-AD mice, with the *p*-value lower than 0.01, 0.05, 0.05, and 0.05. Expressions of sAPPα, post-synaptic density protein 95 (PSD95) and synaptophysin were greatly decreased in 3×Tg-AD mice, with the *p*-values all lower than 0.05. However, in CY-09 treated 3×Tg-AD mice, the expressions of these proteins were reversed, with increased expression of sAPPα, PSD95, and synaptophysin; decreased levels of APP, BACE1, sAPPβ and Aβ1-42 were found in comparison to those in the untreated 3×Tg-AD mice, with the *p*-value lower than 0.05 and 0.01, respectively. Immunostaining of 6E10 confirmed the decrease in the expression and distribution of Aβ in CY-09 treated 3×Tg-AD mice (Figure 8i and Appendix A).

Furthermore, as demonstrated in Figure 9 and Appendix A, we observed increased expression and distribution of pS404-tau in 3×Tg-AD mice, but treatment with CY-09 greatly reduced the expression and distribution, with the *p*-values lower than 0.05 and 0.01. There were no differences in the expressions of Tau5 among the four groups of mice. However, remarkably increased total human tau was found in the 3×Tg-AD mice and the 3×Tg-AD + CY-09 mice, with a *p*-value lower than 0.001, while no reduction was found in the CY-09-treated 3×Tg-AD mice. By contrast, expression of p-GSK3β-Ser9 relative to GSK3β exhibited a decrease in 3×Tg-AD mice (*p* = 0.0834), but it significantly increased after CY-09 treatment, with a *p*-value lower than 0.05. Together, these results demonstrated that CY-09 can reverse the expression and distribution of pathological proteins and alleviate cognitive impairment in the 3×Tg-AD mice.

### 3.8. CY-09 Decreased Oxidative Stress in 3×Tg-AD Mice

Finally, we explored the effects of CY-09 on oxidative stress and ferroptosis. First, we detected ROS and malondialdehyde (MDA) in the four groups of mice. As shown in Figure 10, significantly increased ROS levels and MDA were found in 3×Tg-AD mice, with the *p*-value lower than 0.001 and 0.01. After the CY-09 treatment, ROS and MDA were notably decreased, with the *p*-value lower than 0.001 and 0.05. These data reflected that CY-09 can reduce oxidative stress in 3×Tg-AD mice.

Then, we tested the ferroptosis-related proteins, including TF, TFR, FPN, NCOA4, ACSL4, GPX4, and SLC7A11. TF and TFR were responsible for the transport of Fe^2+^ to the cell, while FPN transported Fe^2+^ outside the cell. NCOA4 participated in the ferritinophagy. GPX4, SLC7A11, and ACSL4 were critical enzymes in lipid peroxidation. As reported in Figure 11, compared with NTg mice, expressions of TFR and ACSL4 were increased, while expressions of FPN, NCOA4, GPX4, and SLC7A11 were decreased in the 3×Tg-AD mice, with the *p*-values all lower than 0.05. Furthermore, when compared with the NTg + CY-09 mice, the expressions of NCOA4, GPX4, and SLC7A11 were significantly decreased in the 3×Tg-AD mice, with the *p*-values all lower than 0.05. However, except for ACSL4, there were no differences in these proteins between the 3×Tg-AD mice and the 3×Tg-AD + CY-09 mice. The expression of ASCL4 was reduced in the 3×Tg-AD + CY-09 mice, with a *p*-value lower than 0.05. No differences in TF expression were found between the four groups of mice except for NTg mice and 3×Tg-AD + CY-09 mice. The results implied that the inactivation of the NLRP3 inflammasome by CY-09 cannot reverse the ferroptosis in the 3×Tg-AD mice.

Reduced expressions of MDA and ACSL4 in the 3×Tg-AD + CY-09 mice prompted us to detect the fatty metabolism changes in the four groups of mice. Data presented in Figure 12, reveal that in comparison to NTg mice, the expressions of acetyl coA carboxylase (ACC), 3-hydroxy-3-methylglutaryl-coenzyme A synthase 1 (HMGCS1), and 3-hydroxy-3-methylglutaryl-coenzyme A reductase (HMGCR) were greatly increased in the 3×Tg-AD mice, with the *p*-values all lower than 0.05. A significant decrease was found in the expression of p-ACC in the 3×Tg-AD mice, with a *p*-value lower than 0.05. After CY-09 treatment, expressions of ACC, p-ACC, HMGCS1, and HMGCR were reversed. Fatty acid synthase (FAS) was also increased in the 3×Tg-AD mice, but with no significant difference in protein levels. Similarly, there were no differences in low-density lipoprotein receptor related protein 1 (LRP1) and sterol regulatory element binding protein 2 (SREBP2) between the four groups of mice, except for LRP1 in CY-09-treated NTg and 3×Tg-AD mice. Increased ACC and FAS and decreased p-ACC indicated increased fatty acid synthesis, consistent with the increased MDA and ACSL4. HMGCS1 and HMGCR are responsible for the synthesis of cholesterol. LRP1 and SREBP2 regulate lipid metabolism homeostasis and cholesterol levels, respectively. The above results suggested that increased fatty acid synthesis may be the main cause of increased lipid peroxidation. CY-09 could reduce the synthesis of fatty acid and lipid peroxidation by inhibiting NLRP3 inflammasome activation in the 3×Tg-AD mice.

## 4. Discussion

Neuroinflammation has been recognized as a key event in inducing the onset and progression of AD, and it is closely linked to Aβ accumulation, tau hyperphosphorylation, and decreased cerebral glucose metabolism [3]. NLRP3 inflammasome is the most important inflammasome involved in neuroinflammation. In this study, CY-09 was used to inhibit the activation of NLRP3 inflammasome, leading to the restoration of glucose metabolism, as demonstrated by the increases in the expression and distribution of glucose transporters and related enzymes and the attenuation of insulin resistance. Moreover, inhibition of NLRP3 inflammasome activation by CY-09 could also reduce oxidative stress and improve cognitive ability in AD model mice.

In recent years, several studies have shown that NLRP3 inflammasome activation contributes to the progress of AD pathologies, specifically Aβ and tau proteins [42,43]. Inhibition of NLRP3 inflammasome activation helps to reduce Aβ deposition and improve cognitive impairment [43]. Most inhibitors of NLRP3 inflammasome activation do not specifically target NLRP3, except for CY-09, which combines directly with the ATP-binding motif of the NLRP3 NACHT domain [32]. Thus, CY-09 was selected in this study to inhibit NLRP3 inflammasome activation. As crossing the BBB and binding to NLRP3 in the brain is the prerequisite for inhibition, we used LC-MS/MS to characterize the molecular structure of CY-09 and detected its level in the brains of the mice. These results proved that CY-09 crossed the BBB to the brain to exert its biological function. Then, we further detected the expression and distribution of Aβ and tau proteins in CY-09-treated and non-treated 3×Tg-AD mice. Consistent with previous studies [43], our results showed reversed expression levels of pathological proteins in CY-09 treated 3×Tg-AD mice. In addition, CY-09 could also relieve cognitive deficits in the AD mice.

Reduced cerebral glucose metabolism has been reported in the AD brain due to the decrease in glucose transport, insulin resistance, and reduced metabolic enzymes [44,45,46]. We used ^18^F-FDG PET to analyze the alteration of glucose metabolism in CY-09-treated and non-treated 3×Tg-AD mice. Our results showed increased glucose uptake and metabolism rates in CY-09-treated 3×Tg-AD mice. GLUT1, GLUT3 and GLUT4 are responsible for glucose uptake and transport in the brain. Decreased expressions of GLUT1 and GLUT3 in AD indicated reduced glucose uptake and transport [47,48,49,50]. In this work, GLUT1 and GLUT3 expression levels were found to be increased in CY-09-treated 3×Tg-AD mice, while GLUT4 also exhibited higher expression and distribution. These results implied that inhibiting NLRP3 inflammasome activation by CY-09 may improve insulin resistance. Insulin is a critical hormone that regulates blood glucose [20,51]. It is transported into the brain and binds with insulin receptors, which are expressed in the cell membranes to activate the insulin signaling pathway [21,52,53]. In AD, insulin levels increase in the blood and decrease in the brain, the insulin signal pathway is suppressed, and GLUT4 cannot localize to the cell membrane, leading to reduced glucose uptake. Intranasal insulin injection helps to recover the insulin signaling pathway, stimulate the transfer of GLUT4 to the cell membranes, and thus increase the uptake of glucose in AD [54]. CY-09 has been reported to inhibit NLRP3 inflammasome activation in order to improve insulin resistance in obesity and non-alcoholic fatty liver disease. In this study, elevated insulin levels and recovered insulin signal pathways were detected in the CY-09-treated AD mice. Increased insulin sensitivity was also measured by GTT and ITT in the CY-09-treated AD mice. We also detected the recovered insulin signal pathway in CY-09-treated 3×Tg-AD mice. These data suggested that the inhibition of NLRP3 inflammasome activation by CY-09 helps to alleviate insulin resistance in AD.

Insulin resistance and glucose metabolism dysfunction are two hallmarks of diabetes mellitus. As they were also found in AD, most researchers considered AD as type 3 diabetes mellitus. However, it should be noted that in diabetes, decreased insulin secretion or insulin resistance leads to increased blood glucose [20,47]. Abnormal insulin level in the blood is the main cause of diabetes. While in AD, the higher insulin levels in the blood and lower levels in the brain indicated a damaged insulin signaling pathway. Therefore, maintaining the stability of insulin levels and blood glucose is the key point for diabetes, but increase the insulin levels and glucose metabolism in the brain is more important in AD.

Glucose was transported into the cells by GLUTs and phosphorylated by mitochondria-bound HK to initiate glucose metabolism. Decreased expression and abnormal distribution of HK were found in AD. Glycolysis and oxidative phosphorylation (OxPhos) are two main pathways of glucose metabolism [55]. Glycolysis generates little ATP, while OxPhos generates a large amount of ATP to meet the energy needs of the neurons [56]. The dividing point between the two pathways is the metabolic selection of pyruvate. Pyruvate is catalyzed by lactate dehydrogenase to generate lactate in glycolysis, while it is catalyzed by PDHE to generate Acetyl-CoA in OxPhos. An increasing number of studies have demonstrated that the metabolic pattern of neurons changes from OxPhos to aerobic glycolysis in the AD brain [25,56,57]. Reduced PDHE expression may be one reason for this shift. Here, we detected the increased expression levels of HK1 and PDHE1α in CY-09 treated 3×Tg-AD mice, which are beneficial for maintaining glucose metabolism and ATP production in the brain.

Oxidative stress is another important pathological characteristic that is closely related to NLRP3 inflammasome and glucose metabolism in AD. ROS was released from the oxidative respiratory chain and can activate the NLRP3 inflammasome [58,59]. Here, we found that the inactivation of the NLRP3 inflammasome by CY-09 decreased ROS levels in CY-09-treated 3×Tg-AD mice. Oxidative stress causes iron metabolism disorders and leads to ferroptosis, which is a new area in the research of AD pathogenesis [25,26]. Ferroptosis manifests as increased cellular Fe^2+^, decreased GPX4, and increased lipid peroxidation [27]. Elevated Fe^2+^ results from increased transferrin transport and ferritinophagy [28]. Consistent with the studies, we detected increased TF, TFR, and FPN and decreased GPX4 and SLC7A11 in the 3×Tg-AD mice. Unfortunately, there was no significant difference in these proteins between the AD and CY-09-treated AD mice. These results indicate that the inactivation of NLRP3 inflammasome does not reduce ferroptosis. Moreover, a contradiction between decreased ferritinophagy and increased ferroptosis was found in AD. Generally, decreased NCOA4 represents decreased ferritinophagy [28,60]. Thus, the expression and function of this protein remain to be clarified.

As a marker of ferroptosis, ACSL4 also participates in lipid peroxidation. Some studies showed that lipid peroxidation is related to Aβ deposition and the hyperphosphorylation of tau [29,30,31]. In this paper, the levels of ACSL4 and MDA were increased in 3×Tg-AD mice and decreased after CY-09 treatment. The results prompted us to explore the effect of CY-09 on lipid metabolism, mainly the synthesis and metabolism of fatty acids and cholesterol. FAS and ACC are rate-limiting enzymes of fatty acid synthesis. In AD mice, the expression of FAS and ACC increased, whereas the phosphorylation level of ACC deceased, thus indicating an increase in fatty acids synthesis. HMGCS1 and HMGCR are important for cholesterol synthesis [61]. Increased HMGCS1 and HMGCR were also found in AD mice. Surprisingly, expressions of ACC, HMGCS1, and HMGCR were decreased in CY-09-treated AD mice. This suggests that decreased MDA and lipid peroxidation may be related to the decreased synthesis of fatty acids and cholesterol in CY-09-treated AD mice. LRP1 plays an important role in lipid metabolism, glucose metabolism, insulin signaling, and the elimination of Aβ in AD [26,62]. Increased LRP1 helps to improve cognitive ability in AD [63]. SREBP2 is a negative regulator of LRP1 [64]. However, in our results, no difference in LRP1 and SREBP2 was found between AD mice and CY-09-treated AD mice. Here, a limitation should be noted. In this study, we used only one strain of AD model mice to study the effect of NLRP3 inflammasome activation on glucose metabolism and the role of CY-09. Using two or more strains of mice would help to validate the experimental results. Therefore, the conclusion drawn in this paper is restricted to only the 3×Tg-AD mice, and it should be verified using additional AD models in the near future.

## 5. Conclusions

Summarily, inhibiting NLRP3 inflammasome activation by CY-09 helps to restore cerebral glucose metabolism, improve memory and learning ability, and reduce fatty acid synthesis and lipid peroxidation in the 3×Tg-AD mice. Thus, CY-09 has the potential to be developed for the treatment of AD. Further studies are required regarding the shift between glycolysis and OxPhos pathways in AD in order to better understand the mechanism of neuroinflammation and glucose metabolism in the development of AD pathology.

## Figures and Tables

**Figure 1 antioxidants-12-00722-f001:**
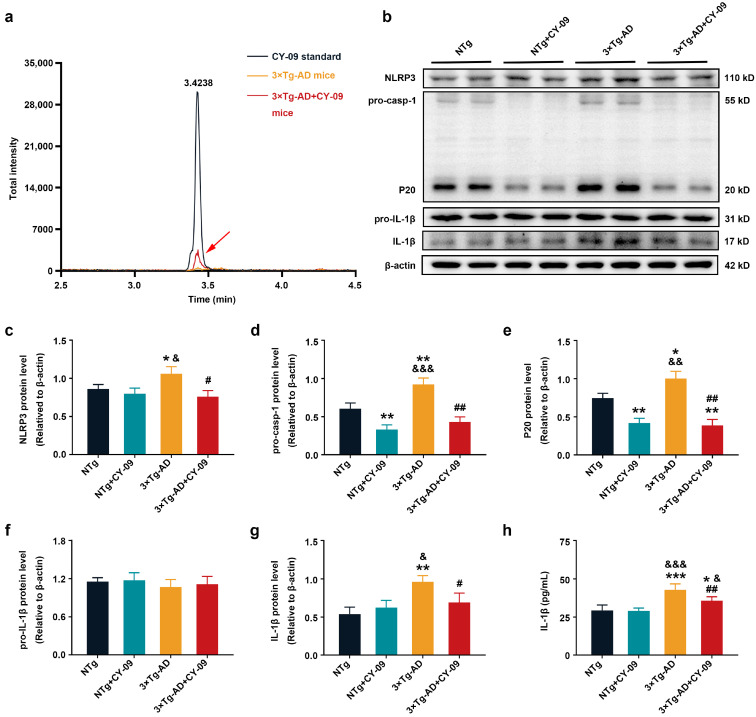
CY-09 attenuated activation of NOD-like receptor protein 3 (NLRP3) inflammasome in triple transgenic AD (3×Tg-AD) mice. (**a**) Content of CY-09 in 3×Tg-AD and 3×Tg-AD + CY-09 mice brains; (**b**–**g**) Western blot analysis of NLRP3, pro-caspase-1, caspase-1 (P20), pro-interleukin-1β (pro-IL-1β), and IL-1β in non-transgenic (NTg), NTg + CY-09, 3×Tg-AD, and 3×Tg-AD + CY-09 mice; (**h**) detection of IL-1β by ELISA in the four groups of mice. (n = 6, mean ± SD, one-way ANOVA and Bonferroni post hoc test; * *p* < 0.05, ** *p* < 0.01, *** *p* < 0.001 vs. NTg mice, ^&^
*p* < 0.05, ^&&^
*p* < 0.01, ^&&&^
*p* < 0.001 vs. NTg + CY-09 mice, # *p* < 0.05, ## *p* < 0.01 vs. 3×Tg-AD mice).

**Figure 2 antioxidants-12-00722-f002:**
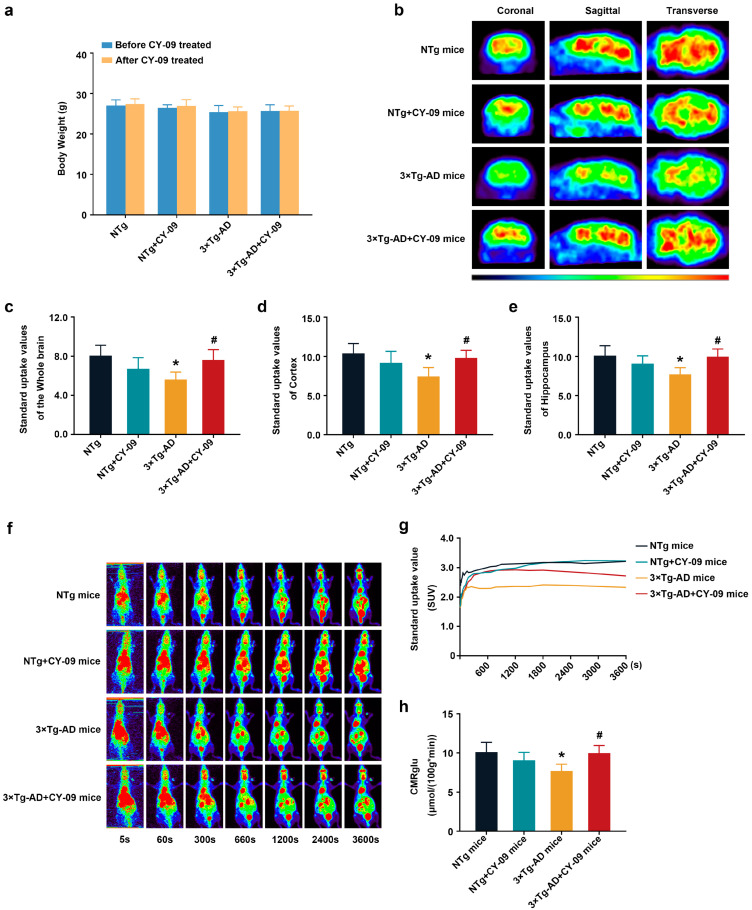
^18^F-FDG positron emission tomography (PET) images of NTg, NTg + CY-09, 3×Tg-AD, and 3×Tg-AD + CY-09 mice. (**a**) Body weight of mice, before and after CY-09 treatment; (**b**) static PET images of the four groups of mice; (**c**–**e**) standard uptake values in the whole brain, cortex, and hippocampus in the four groups of mice; (**f**) dynamic PET images of the four groups of mice; (**g**) cerebral time–activity curves (TAC) and (**h**) cerebral metabolic rate of glucose (CMRglu) in the four groups of mice. (n = 6, mean ± SD, one-way ANOVA and Bonferroni post hoc test; * *p* < 0.05 vs. NTg mice, # *p* < 0.05 vs. 3×Tg-AD mice).

**Figure 3 antioxidants-12-00722-f003:**
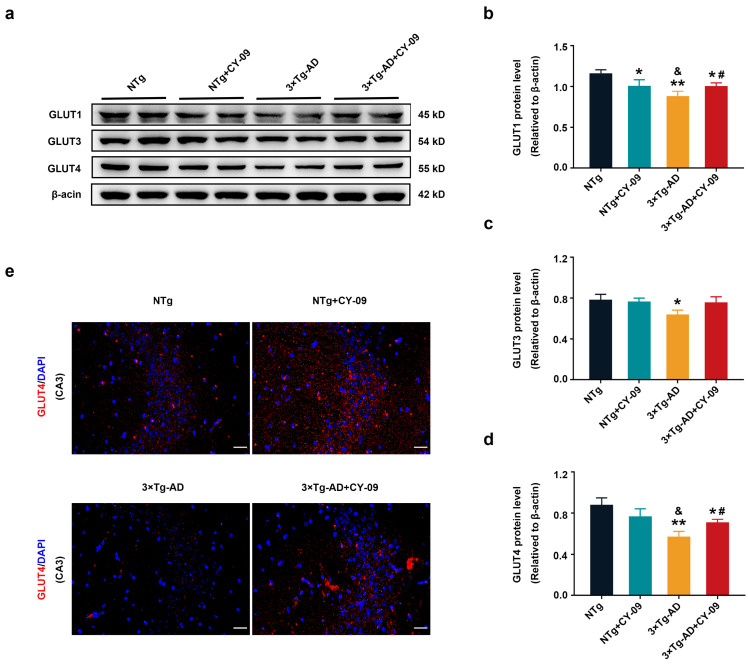
Increased expression of glucose transporters in CY-09-treated 3×Tg-AD mice. (**a**) Representative Western blots and quantification of (**b**) glucose transporters 1 (GLUT1), (**c**) GLUT3, and (**d**) GLUT4 expression in NTg, NTg + CY-09, 3×Tg-AD and 3×Tg-AD + CY-09 mice (n = 6, mean ± SD, one-way ANOVA and Bonferroni post hoc test; * *p* < 0.05, ** *p* < 0.01 vs. NTg mice, ^&^
*p* < 0.05 vs. NTg + CY-09 mice, # *p* < 0.05 vs. 3×Tg-AD mice); (**e**) distribution of GLUT4 in the CA3 region of the four groups of mice (scale bar: 50 μm).

**Figure 4 antioxidants-12-00722-f004:**
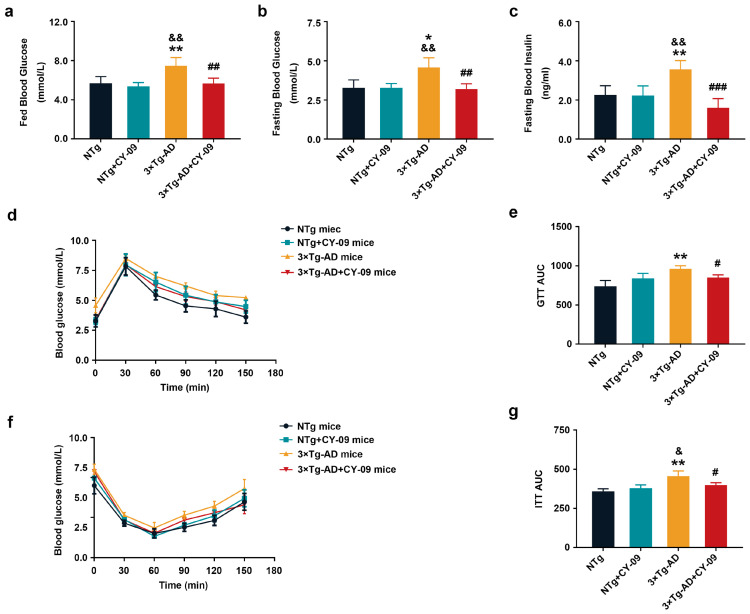
CY-09 relieved the insulin resistance in 3×Tg-AD mice. (**a**–**c**) Fasting blood glucose, fed blood glucose, and fasting blood insulin were measured in NTg, NTg + CY-09, 3×Tg-AD and 3×Tg-AD + CY-09 mice; (**d**,**e**) glucose tolerance tests (GTT) and (**f**,**g**) insulin tolerance tests (ITT) were conducted to determine the insulin sensitivity of the four groups of mice. (n = 6, mean ± SD, one-way ANOVA and Bonferroni post hoc test; * *p* < 0.05, ** *p* < 0.01 vs. NTg mice, ^&^
*p* < 0.05, ^&&^
*p* < 0.01 vs. NTg + CY-09 mice, # *p* < 0.05, ## *p* < 0.01, ### *p* < 0.001 vs. 3×Tg-AD mice).

**Figure 5 antioxidants-12-00722-f005:**
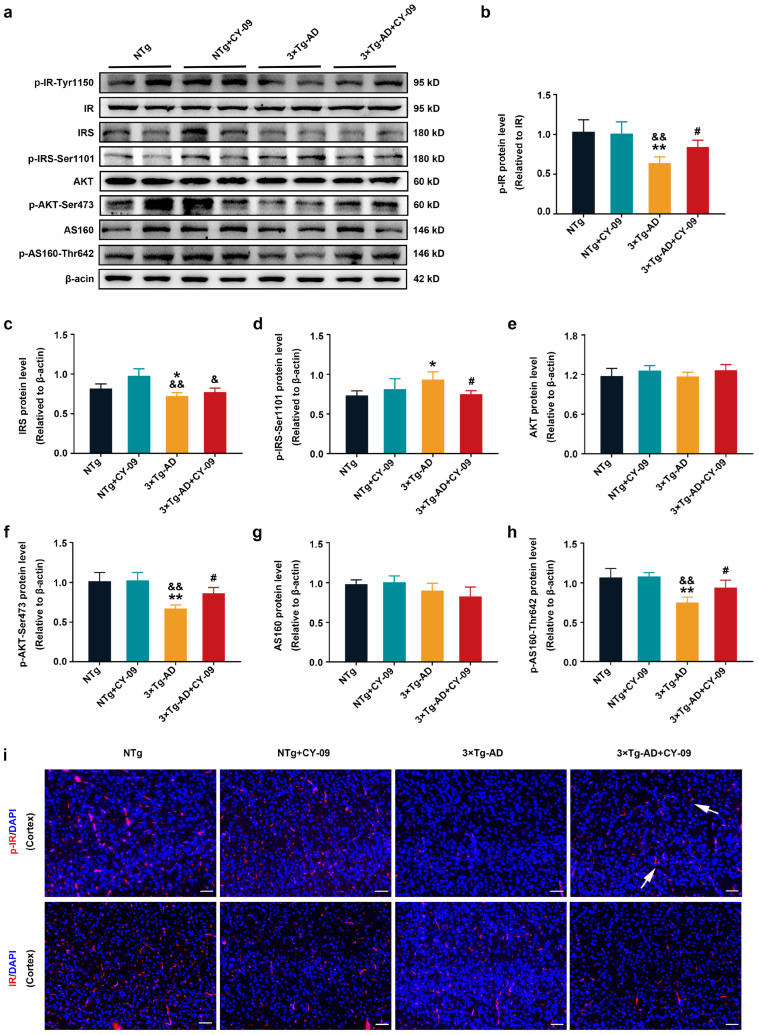
CY-09 restored the insulin signaling pathway in 3×Tg-AD mice. (**a**–**h**) Western blot analysis of insulin receptors (IR), p-IR-Tyr1150, insulin receptor substrate (IRS), p-IRS-Ser1101, AKT, p-AKT-Ser473, AS160 and p-AS160-Thr642 in NTg, NTg + CY-09, 3×Tg-AD, and 3×Tg-AD + CY-09 mice. (n = 6, mean ± SD, one-way ANOVA and Bonferroni post hoc test; * *p* < 0.05, ** *p* < 0.01 vs. NTg mice, ^&^
*p* < 0.05, ^&&^
*p* < 0.01 vs. NTg + CY-09 mice, # *p* < 0.05 vs. 3×Tg-AD mice). (**i**) Distribution of IR and p-IR-Tyr1150 in the brains of the four groups of mice (scale bar: 100 μm; arrows indicate p-IR in 3×Tg-AD + CY-09 mice).

**Figure 6 antioxidants-12-00722-f006:**
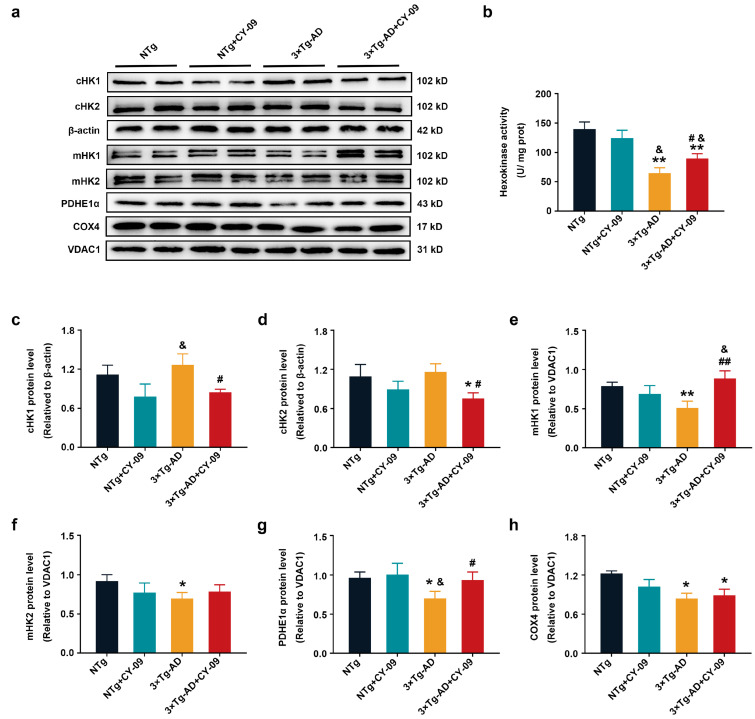
CY-09 reversed the expression and distribution of metabolic enzymes in 3×Tg-AD mice. (**a**,**c**–**h**) Western blot analysis of cHK1 (HK1 in the cytoplasm), cHK2 (HK2 in the cytoplasm), mHK1 (HK1 in mitochondria), mHK2 (HK2 in mitochondria), pyruvate dehydrogenase α 1 (PDHE1α) and cytochrome c oxidase subunit IV (COX4) in NTg, NTg + CY-09, 3×Tg-AD, and 3×Tg-AD + CY-09 mice. (**b**) Detection of hexokinase activity in the brain tissue of the four groups of mice. (n = 6, mean ± SD, one-way ANOVA and Bonferroni post hoc test; * *p* < 0.05, ** *p* < 0.01 vs. NTg mice, ^&^
*p* < 0.05 vs. NTg + CY-09 mice, # *p* < 0.05, ## *p* < 0.01 vs. 3×Tg-AD mice).

**Figure 7 antioxidants-12-00722-f007:**
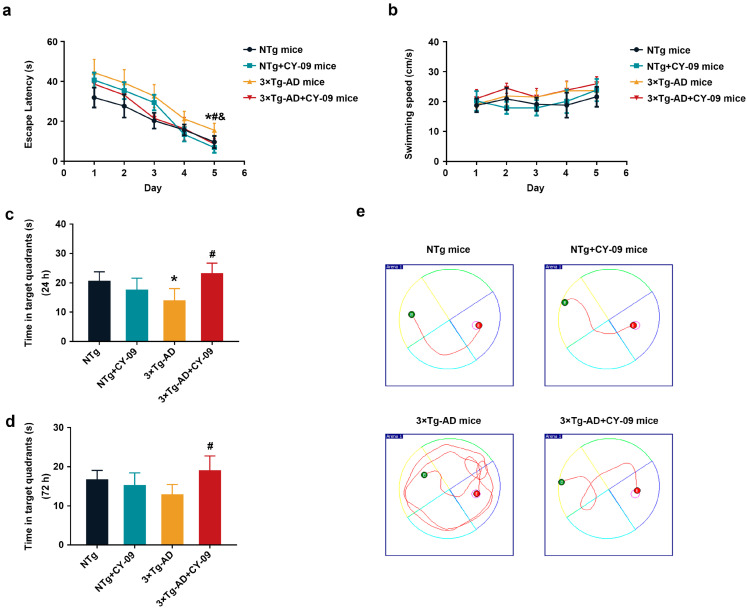
CY-09 improved the learning and memory ability of 3×Tg-AD mice. (**a**) Escape latency and (**b**) swimming speed during training of the four groups of mice. (n = 6, mean ± SD, one-way ANOVA and Bonferroni post hoc test; * *p* < 0.05: 3×Tg-AD mice vs. NTg mice, ^&^
*p* < 0.05: 3×Tg-AD mice vs. NTg + CY-09 mice, # *p* < 0.05: 3×Tg-AD + CY-09 mice vs. 3×Tg-AD mice). (**c**,**d**) Time in target quadrants in 24 h and 72 h space exploration experiments. (n = 6, mean ± SD, one-way ANOVA and Bonferroni post hoc test; * *p* < 0.05 vs. NTg mice, # *p* < 0.05 vs. 3×Tg-AD mice). (**e**) Schematic diagram of the swimming paths of the four groups of mice.

**Figure 8 antioxidants-12-00722-f008:**
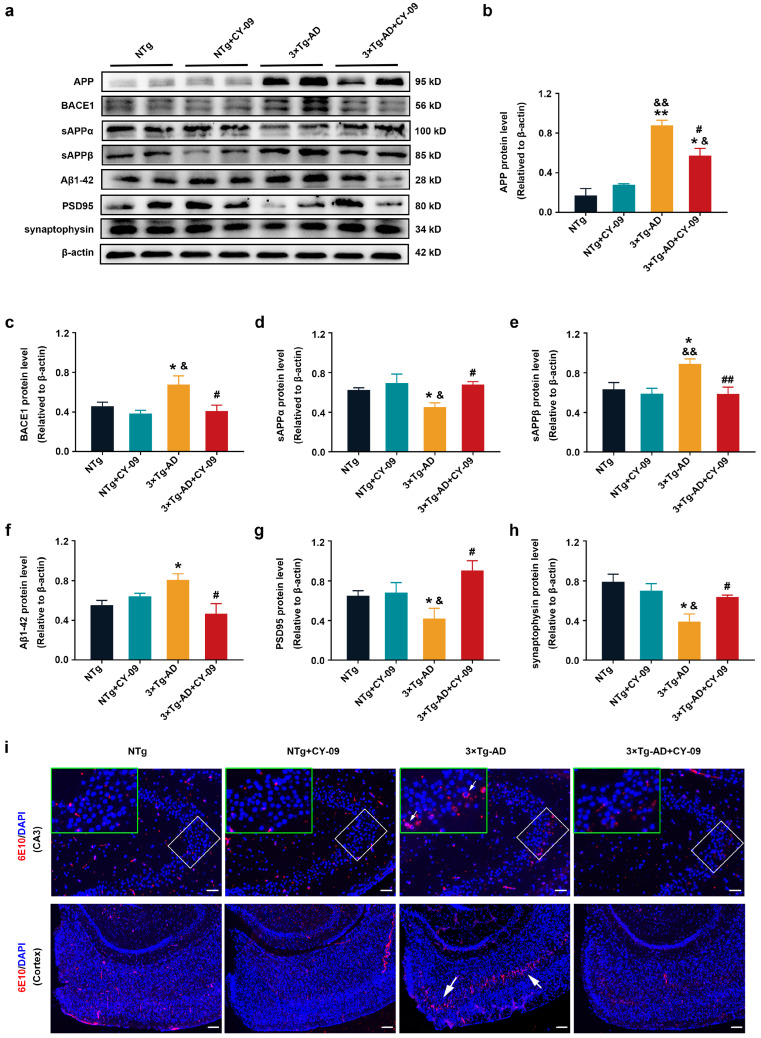
CY-09 reversed the expression and distribution of Aβ related proteins in 3×Tg-AD mice. (**a**–**h**) Western blot analysis of amyloid precursor protein (APP), beta-site app cleaving enzyme 1 (BACE1), sAPPα, sAPPβ, Aβ1-42, post-synaptic density protein 95 (PSD95), and synaptophysin in NTg, NTg + CY-09, 3×Tg-AD, and 3×Tg-AD + CY-09 mice. (n = 6, mean ± SD, one-way ANOVA and Bonferroni post hoc test; * *p* < 0.05, ** *p* < 0.01 vs. NTg mice, ^&^
*p* < 0.05, ^&&^
*p* < 0.01 vs. NTg + CY-09 mice, # *p* < 0.05, ## *p* < 0.01 vs. 3×Tg-AD mice). (**i**) Distribution of Aβ in the CA3 region and cortex of the four groups of mice (scale bar: 100 μm and 250 μm; arrows indicate Aβ in 3×Tg-AD mice).

**Figure 9 antioxidants-12-00722-f009:**
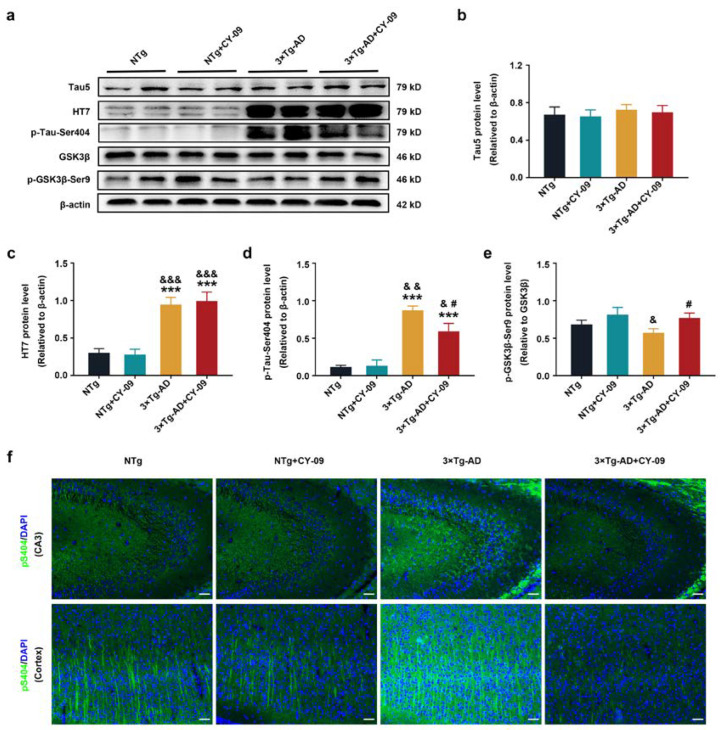
CY-09 reversed the expression and distribution of tau-related proteins in 3×Tg-AD mice. (**a**–**e**) Western blot analysis of Tau5, HT7, p-Tau-Ser404, GSK3β, and p-GSK3β-Ser9 in NTg, NTg+CY-09, 3×Tg-AD, and 3×Tg-AD + CY-09 mice. (n = 6, mean ± SD, one-way ANOVA and Bonferroni post hoc test; *** *p* < 0.001 vs. NTg mice, ^&^
*p* < 0.05, ^&&^
*p* < 0.01, ^&&&^
*p* < 0.001 vs. NTg + CY-09 mice, # *p* < 0.05 vs. 3×Tg-AD mice). (**f**) Distribution of p-Tau-Ser404 in the CA3 region and cortex of the four groups of mice (scale bar: 100 μm).

**Figure 10 antioxidants-12-00722-f010:**
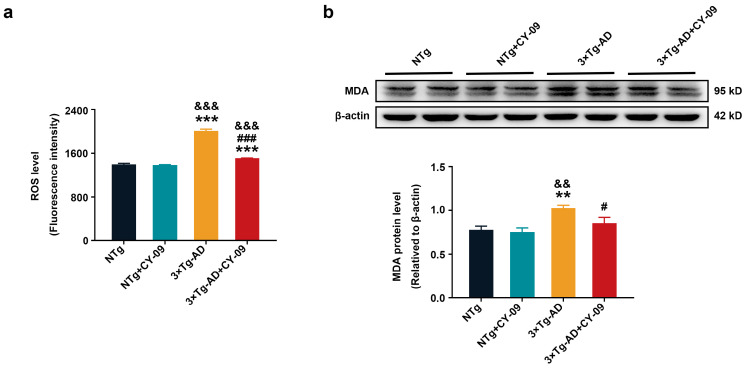
CY-09 decreased reactive oxygen species (ROS) levels and malondialdehyde (MDA) expression in the four groups of mice. (**a**) ROS levels in NTg, NTg + CY-09, 3×Tg-AD, and 3×Tg-AD + CY-09 mice; (**b**) expression of MDA in the four groups of mice. (n = 6, mean ± SD, one-way ANOVA and Bonferroni post hoc test; ** *p* < 0.01, *** *p* < 0.001 vs. NTg mice, ^&&^
*p* < 0.01, ^&&&^
*p* < 0.001 vs. NTg + CY-09 mice, # *p* < 0.05, ### *p* < 0.001 vs. 3×Tg-AD mice).

**Figure 11 antioxidants-12-00722-f011:**
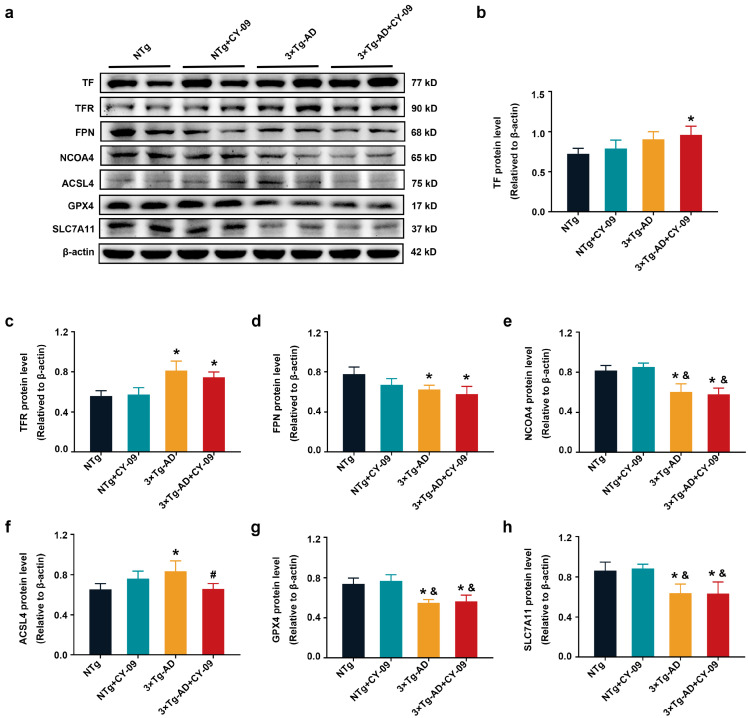
CY-09 could not reverse the ferroptosis in 3×Tg-AD mice. (**a**–**h**) Western blot analysis of transferrin (TF), transferrin receptor (TFR), ferroportin (FPN), nuclear receptor coactivator 4 (NCOA4), long-chain acyl-CoA synthetase 4 (ACSL4), glutathione peroxidase 4 (GPX4), Solute Carrier Family 7, and Member 11 (SLC7A11) in the four groups of mice. (n = 6, mean ± SD, one-way ANOVA and Bonferroni post hoc test; * *p* < 0.05 vs. NTg mice, ^&^
*p* < 0.05 vs. NTg + CY-09 mice, # *p* < 0.05 vs. 3×Tg-AD mice).

**Figure 12 antioxidants-12-00722-f012:**
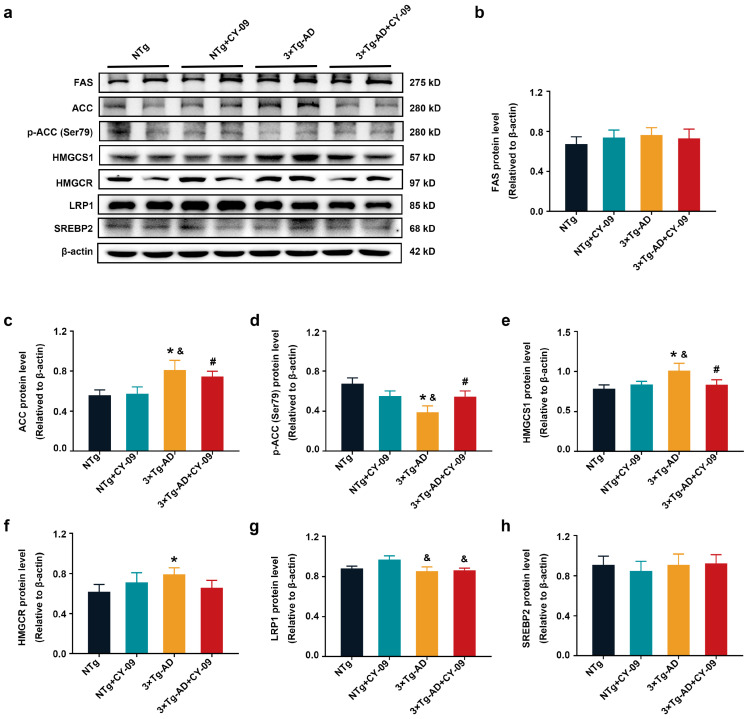
CY-09 decreased the synthesis of fatty acids in 3×Tg-AD mice. (**a**–**h**) Western blot analysis of fatty acid synthase (FAS), acetyl CoA carboxylase (ACC), p-ACC, 3-hydroxy-3-methylglutaryl-Coenzyme A synthase 1 (HMGCS1), 3-hydroxy-3-methylglutaryl-Coenzyme A reductase (HMGCR), low-density lipoprotein receptor related protein 1 (LRP1), and sterol regulatory element binding protein2 (SREBP2) in the four groups of mice. (n = 6, mean ± SD, one-way ANOVA and Bonferroni post hoc test; * *p* < 0.05 vs. NTg mice, ^&^
*p* < 0.05 vs. NTg + CY-09 mice, # *p* < 0.05 vs. 3×Tg-AD mice).

## Data Availability

The data presented in this study are available on request from the corresponding author.

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
