# Peer review of "Inhibiting NLRP3 Inflammasome Activation by CY-09 Helps to Restore Cerebral Glucose Metabolism in 3×Tg-AD Mice"

_antioxidants, 2023, doi:10.3390/antiox12030722_

Round 1
Reviewer 1 Report
Review comments are attached

Reviewer 2 Report
paper titled "Inhibiting NLRP3 Inflammasome Activation by CY-09 Helps to Restore Cerebral Glucose Metabolism in Alzheimer’s Disease" by Han et al. studied the role of CY-09 " an NLRP3 inhibitor" in restoring cererbral glucose metabolism in an animal model of Alzheimer's disease.
1- Title : needs to show that the study was done in animal model
2- Abstract should be amended by some numerical values
3- Key words: do not include the animal model
4- Line 23: and inactivation of NLRP3 inflam-: better to say "inhibition" instead of inactivation, it does not sound well after the first senetnce.
5- The rationality of the study is not strong: the title included the role of NLRP3 on glucose metabolism, however, in the intorduction, authors did not succeed to convince me that there is arational or relation between NLRP3 inhibition and glucose metabolism
6- What is the weight of the animals at the begin of the study? Why authors selected female mice (not male ) for this study?
7- What were the experiemntal groups? how many group and what treatments ? this is not mentioned in the methods!!!
8- Methods in general lacks references for example: MWM test.
9- in MWM: were the mice trained on the test before experiemntation?
10- WB analysis: how samples were kept before performing the WB assay?
11- How we moved form behavioural tasks & GTT to ELISA assays? how were the animals sacrificed ? how blood and organs were collected and kept?
12- Authors have to check the normality of distribution of the results by a suitable post hoc test (such as Shapiro-Wilk test or K-S test) before deciding to choose certain ANOVA. If the normality test indicated normal dist of the data, so use one-way ANOVA, if not, use non parametric ANOVA. In all cases choose a suitable post-hoc test
13- Authors should give the source of chemicals, kits and antibodies completely and consistently (code, company, town, state and country) & version for software.
14- Authors need to show every possible comparison between the study groups by post hoc analysis
Reviewer 3 Report
The present manuscript was well organized. There were some minor comments to improve the manuscript.
1. In Introduction, the authors should add the background about ferroptosis and oxidative stress in Alzheimer's disease (AD) assessed and written in Results. Moreover, the authors wrote reduction of glucose metabolism was reported to be related to NLRP3 inflammasome activation in Abstract; however, it was not written in Introduction. It was written that glucose metabolism and NLRP3 activation in AD pathology and inhibition of NLRP3 improving the AD pathology; however, whether changes in glucose metabolism affected NLRP3 activation was not explained in Introduction.
2. In the present study, the authors used female mice. Why?
3. Results 3.1 and Figure 1a, was not CY-09 found in the brain of NTg+CY-09 mice? In the graph of Figure 1a, lines with NTg and NTg+CY-09 were not shown.
4. The image in Figure 1b did not match to the graph in Figure 1c. The image should be more representative one. Figure 1d and 1e looked that NTg+CY-09 decreased each protein level compared to NTg. Were there any significant differences?
5. In the images of immunofluorescent staining in Figure 3e, 5i, 8i, and 9f, the explanations of scale bars and arrow heads should be added in figure legends. In addition, the quantitative analysis for the images should be added to show in text the changes of expression levels.
Round 2
Reviewer 2 Report
The revised form of paper titled (Inhibiting NLRP3 Inflammasome Activation by CY-09 Helps to Restore Cerebral Glucose Metabolism in 3×Tg-AD Mice) is very partly improved & many questions are not answered approriately & some answers are only hidden in the reply letter -although not sufficient- and did not appear inside the manuscript. Methods still needs references as well. Actually this article is not suitable for publication in Antioxidants.
1- The new paragraph in introduction needs revision for English grammar.
2- Authors did not mention the rational of using female mice inside the maunscript.
3- Authors did not rationalize using 2 different strains of mice to validate the experiemntal design
4- "in MWM: were the mice trained on the test before experimentation?" : this question was answered only in the reply letter & did not appear inside the paper.
5- The WB procedures should be written in details for steps & detailed origin of every antibody, kit or instrument/software for quantification.
6- 12- Authors have to check the normality of distribution of the results by a suitable post hoc test (such as Shapiro-Wilk test or K-S test) before deciding to choose certain ANOVA. I: this question in the original revision was not answered inside the maunscript
7- Authors should give the source of chemicals, kits and antibodies completely and consistently (code, company, town, state and country) & version for software.: answer this in full please
8- Authors need to show every possible comparison between the study groups by post hoc analysis: authors did not confirm on this in the original manuscript
Round 3
Reviewer 2 Report
The revised version 2 of paper titled (Inhibiting NLRP3 Inflammasome Activation by CY-09 Helps to Restore Cerebral Glucose Metabolism in Alzheimer’s Disease) by Han et al. was not perfectly revised.
Post hoc Statistical analysis should be revised completely and confirm in methods "every possible comparison was explored " & in figures authors should use more clear format to refer to signifcant differences between 2 grouops by using standard sympols Rather than the lines as they the reader will not be sure what this line referes to. This is mandatory to validate the results of the study & hence the conclusion drawn from it.
Introduction also has some unnecssary statements and can be shortened to be more concrete
